# Development and Characterization of PA 450 and PA 3282 Epoxy Coatings as Anti-Corrosion Materials for Offshore Applications

**DOI:** 10.3390/ma15072562

**Published:** 2022-03-31

**Authors:** Mohammad Asif Alam, Ubair Abdus Samad, Asiful Seikh, Jabair Ali Mohammed, Saeed M. Al-Zahrani, El-Sayed M. Sherif

**Affiliations:** 1Center of Excellence for Research in Engineering Materials (CEREM), Deanship of Scientific Research, King Saud University, P.O. Box 800, Riyadh 11421, Saudi Arabia; moalam@ksu.edu.sa (M.A.A.); aseikh@ksu.edu.sa (A.S.); jmohammed@ksu.edu.sa (J.A.M.); 2Department of Chemical Engineering, College of Engineering, King Saud University, P.O. Box 800, Riyadh 11421, Saudi Arabia; szahrani@ksu.edu.sa

**Keywords:** polyaminoamine and polyamidoamine adducts, epoxy coatings, corrosion resistance, nanoindentation, electrochemical impedance spectroscopy

## Abstract

The optimization of two different types of hardeners, namely polyaminoamine adduct (Aradur 450 BD) and polyamidoamine adduct (Aradur 3282 BD), with diglycidyle ether of bisphenol-A (DGEBA) epoxy resin was carried out. Three different stoichiometries of PA 450 to the epoxy resin to fabricate E-0, E-1, and E-2 coating samples and the other three of PA 3282 to the epoxy resin to fabricate F-0, F-1, and F-2 coating samples were coated on mild steel panels. All coated samples were characterized by scanning electron microscopy (FE-SEM), Fourier transform infrared spectroscopy (FTIR), thermo-gravimetric analysis (TGA), and nanoindentation techniques. The electrochemical corrosion behavior of the fabricated coatings was investigated using electrochemical impedance spectroscopy (EIS) after various exposures in the climatic conditions in 3.5% NaCl solutions. It was found that the coatings possess almost identical thermal and mechanical properties. Moreover, the E-1 coating shows better corrosion resistance compared to E-0 and E-2 coatings. On the other hand, the F-1 coating was the most effective in significantly improving corrosion resistance. Overall, the addition of PA 450 and PA 3282 to some stoichiometries improves the corrosion resistance of the fabricated coatings.

## 1. Introduction

Metals are abundantly used in various industrial applications because they offer tremendous mechanical properties. However, their resistance against corrosion in a natural habitat without any protection is of major concern. Corrosion is a naturally occurring phenomenon that leads to considerable financial losses in different areas of structural importance [1]. In order to protect the metals from corrosion, the most common practice around the world is to coat metallic surfaces with coatings to reduce or delay the process of corrosion [2]. There are different mechanisms by which metals can be protected that include (a) barrier coating (prevent corrosive species from reaching the metal), (b) noble metal coating (to ensure prevention of corrosion of the base metal), (c) sacrificial metal coating, (d) inhibitor coatings, and (e) electrical resistive coating (organic coating) [3,4,5]. In today’s industrial applications, polymeric coatings are the ones that are mainly utilized for corrosion prevention applications [6,7,8]. These coatings prevent corrosion by acting as a barrier against corrosive species. Coatings not only decrease degradation, but also protect metallic substrates from various other factors such as humidity, UV radiation, and mechanical abuse. [3,4,5,9,10,11,12].

With the availability of different coating systems such as vinyl-based coatings, polyester coatings, acrylic coatings, epoxy, polyurethane, etc., the use of epoxy resin is very common in outdoor applications because of the advantages it offers. These are widely used in many industrial applications, especially petrochemical, aeronautic, and automotive [13,14,15]. Epoxy resins are chosen because of their mechanical strength, excellent adhesion, corrosion and chemical resistance, and thermal stability [16]. These properties could be attributed to the higher cross-linked density of epoxy due to the presence of amines (NH_2_) and hydroxyl groups in the polymer coating system [17]. Similar to the type of resin chosen for making durable coatings, the type of curing agent also affects the final properties. With the availability of many types of curing agents, polyamine, polyamide, and polyamidoamines are the ones mostly used. These curing agents are used due to their ease in processing at room temperature. However, the final properties and the performance of an epoxy coating are dependent on both the molecular weight of the epoxy resin and the curing agent’s type as well as concentration [18,19,20,21].

The coatings prepared using epoxy resin are brittle because of their higher cross-linked density [22]. These coatings, over a small period of time, produce cracks. These defects then propagate because of epoxy’s brittle nature. These defects in the coating then assist the migration of corrosive ions which ultimately reach the coating/metal interface, causing localized corrosion which leads to delamination [23,24]. These defects can be minimized with the optimization of proper epoxy resin and hardener concentration in order to achieve the best set of properties a coating can offer prior to the addition of any filler materials. There are some studies reported in the literature on mixing ratios of hardener and epoxy resin to analyze the effect of resin and hardener quantities higher than stoichiometric quantities. Wu et al. [25] analyzed the effect of epoxy and hardener mix rationing on water absorption, crosslinking network formation, and dynamic modulus. They reported that the epoxy curing rate increases with increasing hardener amount and that with an excess of hardener, the epoxy linkages are loosely bonded. In wet environments, the excess of hardener accelerates water absorption. Density measurement with various hardener compositions shows different values of moisture absorption, which was mainly because of differences in epoxy network structure. They found that optimal ratio in a moisturized environment was much lesser than stoichiometric quantity. Vanlandingham et al. [26] also reported the relation between stoichiometry and the properties of amine-cured epoxy. They proposed the presence of soft and harder phases in the system, which tends to change with the ratio of these phases. These two phases exhibit a single Tg, but this Tg depends on cross-linked density. They also reported an increase in fracture toughness with the increase of amine content, which happens because of an increased soft phase. In our previous studies [27,28] we also reported the effect of changing epoxy to hardener ratio on various properties of the coating. According to those findings, the optimal accepted ratio where acceptable mechanical and electrochemical properties were obtained was different from the stoichiometric ratio.

In this current study, we optimized the two different types of epoxy hardeners, namely polyaminoamine adduct Aradur-PA450 and Aradur-PA3282, with different hardener concentrations in order to find the best combination with good mechanical and anticorrosion properties. These coatings were characterized with the help of a scratch test and impact test following ASTM standards. Nanomechanical characterization was also performed with the help of a nanoindenter test. Anticorrosion studies of the optimized coatings were performed by immersing the coatings after curing in 3.5% NaCl solution, and the analysis was conducted at different exposure periods from 1 h to 30 days.

## 2. Materials and Methods

The procurement of epoxy resin of DGEBA (diglycidyl ether bisphenol A) was done from Hexion chemicals (Iserlohn, Germany). Both types of hardeners (Aradur-450 and Aradur-3282) were purchased from Huntsman Advanced Materials (Bergkamen, Germany). The solvents used to facilitate the preparation of coatings such as acetone, xylene, and methyl isobutyl ketone (MIBK) were purchased from the local Saudi Arabian market (Ideal Chemicals, Riyadh, Saudi Arabia). All materials were utilized as received without any changes or modifications.

The coatings’ formulations were prepared by diluting the epoxy resin with the help of solvents to obtain a particular viscosity in order to ease the application of coating on the substrate. After the dilution of the epoxy resin, the hardeners were mixed with the resin in different stoichiometric ratios. After mixing, the prepared formulations were left for stabilization for 10 min. After stabilization, the formulations were applied to substrates with the help of an automatic applicator (Sheen, Surrey, UK) and left for 7 days to completely cure the coatings. Formulating ingredients such as epoxy resin and hardener were originally mixed in the stoichiometric ratio of 5:1 (epoxy to hardener). To obtain the best set of properties, apart from the original mixture, two other coating formulations were prepared for each hardener with a variation in hardener percentage of ±5%, which is described in Table 1 below.

After 7 days, the coated panels were analyzed to measure the properties of prepared coatings in order to find the best combination of mechanical and electrochemical properties. The coatings were subjected to FTIR (Fourier-transform infrared spectroscopy) measurements to analyze the changes in bonding and coating chemistry with the variation of hardener percentages. The morphology of coatings was checked with the help of SEM (scanning electron microscope, Joel, Tokyo, Japan). The mechanical properties of the prepared coatings with different hardener percentages were analyzed with the help of conventional testing techniques such as pendulum hardness (ASTM D-4366), impact resistance (ASTM D-2794), and scratch testing (ASTM D-7027). The pendulum hardness (Koenig pendulum tester: model 707/K, Sheen, Surrey, UK) was used to define the surface hardness of coatings by measuring the number of oscillations on the coatings’ surfaces; higher oscillations corresponded to higher surface hardness. The impact strength (Gardner impact tester: model IG-1120, BYK, Columbia, SC, USA) was measured by dropping a standard load on the surface of coating from different heights. The height at which the coating ruptures is taken as impact failure. The scratch resistance (scratch tester: model 705, Sheen, Surrey, UK) was measured by increasing the load against a movable mounting bed where samples are mounted. The load is gradually increased from 500 g to a maximum of 10 kg. The weight at which the coating ruptures is taken as the failure weight. The nanomechanical properties of the coatings were analyzed with the help of nano test platform 3 from micromaterials. A Berkovich (Micromaterials, Wrexham, UK) type indenter was used to analyze the coatings properties using load control program. The coatings were subjected to maximum load of 250 mN with loading rate of 1 mN/s. Upon reaching the maximum load, the load was held for 60 s to remove anomalies related to creep. After that, the load was completely removed at the same rate of 1 mN/s. At least 5 indentations were taken on each sample and different locations and the results are provided as averaged. Electrochemical impedance spectroscopy (EIS) was performed with the help of Autolab Ecochemie PGSTAT 30 (Metrohm, Amsterdam, The Netherlands) using a conventional 3-electrode cell. The coatings were exposed to a 3.5% NaCl solution prior testing, the coatings were exposed to different time intervals from 1 h to a maximum of 30 days.

## 3. Results and Discussion

### 3.1. Fourier-Transform Infrared Spectroscopy (FTIR)

The bond formation and functional group analysis for the formulated coatings, which were prepared with a variable percentage of hardener, were analyzed with the help of FTIR. The analysis was carried out on a thin film through which light can pass easily in order to analyze the functional groups formed due to epoxy and hardener reactions and the changes associated with the variation in the hardener percentage (see Table 1). The FTIR spectra shown in Figure 1 and Figure 2 were acquired after complete curing on a dry surface. The spectra were collected at a wavenumber range between 400 cm^−1^ and 4000 cm^−1^ to see the transmittance for every spectrum apparent on the coated surfaces.

The chemical structure of the DGEBA molecule contains variety of organic groups such as aromatic rings, -N=N-, -NH_2_, -OH, =C=O, -CH_3_, ≡C-O-C≡, etc. It has also been reported [29] that the polyamide hardener molecule (PA450 and PA-3282) contains multi-functional groups of -OH, -NH_2_, -N=N-, =C=O, etc. A combination of DGEBA and these hardeners with stoichiometric variations shown in Table 1 would probably lead to a compound that contains all of their organic and aliphatic groups. It is clearly seen from Figure 1 and Figure 2 that, all spectra showed the same peaks: 768, 827, 945, 1037, 1107, 1182, 1239, 1295, 1392, 1459, 1508, 1606, 1704, 2361, 2853, 2927, and 3336 cm^−1^. The appearance of the peaks at 1606, 1508, and 1392 cm^−1^, as well as those at 1000–1400 cm^−1^ resulted from aromatic rings [30,31]. Also, the C-H out-of-plane deformation vibration bands at 827 and 730–770 cm^−1^ are from the ring vibrations. The broad band at 3336 cm^−1^ is mainly due to the O-H stretching of hydroxyl groups and might form due to the formation of DGEBA-PA epoxy coating [31]. The only difference that is identified from the FTIR spectra for the different epoxy coatings which can also be seen in figures is that the increase of the stoichiometric variation increases the intensity of the light transmittance of the coating.

### 3.2. Field Emission Scanning Electron Microscope (FE-SEM)

The samples were characterized under a scanning electron microscope. The samples were mounted on stubs. A thin layer of platinum was applied using a spurting coater (Joel, Tokyo, Japan) to avoid charging of the samples. The obtained SEM images for prepared samples are shown in Figure 3. It can be seen in Figure 3 that all the coatings possess very smooth surfaces.

### 3.3. Thermogravimetric Analysis (TGA)

In order to analyze the effect of hardener variation on thermal properties, the coatings were subjected to thermogravimetric analysis (TGA). All the samples were heated using a ramp control program with a ramp rate of 10 °C per minute from room temperature up to 600 °C under nitrogen flow. The obtained graphs are shown in Figure 4, while Table 2 shows the temperature at specific weight loss percentages.

It can be seen in figures that a similar weight loss pattern was observed in the graphs for both type of hardener with initial degradation starting from 100 °C and major phase of decomposition; that is, the burning/breaking of main chain started above 300 °C. The initial degradation is related to the removal of moisture trapped in the coatings because the curing was performed at room temperature for 7 days. This is likely due to the decomposition of residual reactants or the decomposition of low molecular weight fractions [32,33].

The second and major decomposition started above 300 °C, which is related to the degradation of the main epoxy cross-linked network [34]. As epoxy is thermoset, it decomposes directly rather than melting. This degradation is rapid because of main backbone chain degradation. It is worth noting that PA-450 gives better stability and higher degradation temperatures at 25% and 50% decomposition, while at 75% decomposition the obtained temperature ranges are the same.

### 3.4. Mechanical Properties and Nanoindentation

The mechanical properties of the coatings were analyzed to check the influence of change in hardener percentages. The coatings were characterized with the help of conventional testing techniques such as pendulum hardness, scratch, and impact. The results obtained from this testing are summarized in Table 3 for both hardeners (PA-450 and PA-3282).

The results presented in Table 3 and Figure 5 represent the coatings’ bulk properties, which suggest that there is no noticeable change in mechanical properties when comparing with the change in hardener percentage because these coatings are based on very little change in hardener percentage without any filler addition. On the other hand, the properties obtained with PA-450 tend to perform slightly better in terms of mechanical properties in comparison with the properties obtained with PA-3282. It can be seen that, with different percentages of hardener PA-450, slightly higher pendulum hardness and impact strength are observed. This is because of the higher cross-linked density achieved by PA-450 with the epoxy resin, which reflects directly in the results obtained for pendulum hardness and impact strength [35]. The same coatings were subjected to nanoindentation characterization to analyze the hardness and modulus of the coatings. Figure 4 and Figure 5 below represent the load vs. depth profile for the coatings prepared with PA-450 and PA-3282, respectively.

It can be seen in Figure 6 and Figure 7 that the loading process of the coatings was very smooth with no abrupt depth penetration while load was increased. All the coatings withstood until a maximum load of 250 mN was achieved. It is worth noting that balanced stoichiometric formulation (refer Table 1) in both the hardener cases, i.e., PA-450 and PA-3282, are the formulations with higher resistances to penetration. Analysis of these graphs gives us an idea about the possible changes in hardness and modulus with changes in hardener concentration during stoichiometric balance.

The following test performed on coating samples was analyzed with the help of software provided by micromaterials, which works on the improved version of Oliver and Pharr theory [36]. According to their model, the values of hardness and elastic modulus are derived using following Equations (1) and (2):H = F_max_/A(1)
E_r_ = 1 − ʋ^2^/E + 1 − ʋ_i_^2^/E_i_(2)
where H represents hardness, A is the projected area at maximum load, and F_max_ is the maximum applied load. E is the modulus of samples, E_r_ is the reduced modulus (obtained from test), ʋ is the poisons ratio (0.35 for polymers), ʋ_i_ is the indenter poisons ratio (0.07 for diamond indenter), while E_i_ is the modulus of diamond indenter (1140 GPa). The results obtained from the analysis are presented in Table 4 below and their graphical representation is given in Figure 8.

The obtained results presented in Table 4 suggest that the coatings prepared with PA-450 possess better hardness and modulus compared to the coatings prepared with PA-3282. This is because of the better crosslink ability of PA-450 hardener. The obtained indentation results are in accordance with the results obtained with conventional mechanical analysis, whereas pendulum hardness (indication of surface hardness) and impact were higher with this type of hardener.

### 3.5. Electrochemical Impedance Spectroscopy (EIS)

The corrosion behavior of the epoxy coatings was reported by the use of the non-traditional electrochemical impedance spectroscopy (EIS) technique, which was successfully employed in similar studies [27,37]. These measurements were performed in a conventional three-electrode cell that has Ag/AgCl (in a saturated KCl solution) as a reference electrode, stainless steel as a counter electrode, and steel-epoxy coated coupons as working electrodes. The EIS data were collected after varied exposure periods of time; 1 h, 7 days, 14 days, 21 days, and 30 days in 3.5% NaCl solutions. The EIS experiments were collected using an Autolab Ecochemie PGSTAT 30 (Metrohm, Amsterdam, The Netherlands). The frequency scan within the range of 0.1 to 100,000 Hz was applied. These EIS experiments were carried out by applying a ±5 mV amplitude sinusoidal wave perturbation to the corrosion potential.

The obtained Nyquist plots for the coating prepared with different Aradur 450 BD (PA 450) percentages (E-0, E-1 and E-2) after 1 h immersion in 3.5% NaCl solution are shown in Figure 9. The same measurements were acquired after immersing the same coated coupons (E-0, E-1, and E-2) in the sodium chloride solution (3.5% NaCl) for varied exposure periods. The Nyquist plots that were obtained after immersing the different coupons for 7 days, 14 days, 21 days, and 30 days are displayed in Figure 10, Figure 11, Figure 12, and Figure 13, respectively.

The different collected EIS data have been fitted to an equivalent circuit that represents the best fit and are shown in Figure 14. The values of the elements of this circuit are listed in Table 5. These elements are the solution resistance, R_S_; the polarization resistance, R_P1_; the constant phase elements, Q_1_; another polarization resistance, R_P2_; and another constant phase element, Q_2_. Here, the first polarization resistance, R_P1_, is the resistance between the interface of the solution and the outer layer of the epoxy coating, while the second polarization resistance (R_P2_) is considered as the corrosion resistance for the interface between the top formed layer (oxide and/or corrosion product’s layer) and the solution [38,39,40,41]. Moreover, the total polarization resistance (R_PT_) for the coatings is obtained by gathering both R_P1_ and R_P2_; R_PT_ = R_P1_ + R_P2_. J. Mayne [42] investigated the mechanism of the inhibition of the corrosion of iron and steel by means of paint and reported that the value of R_PT_ expresses the overall resistance to ion transport through the coating. They also reported that this is the most important factor used to determine the anticorrosive protection that can be obtained by the use of the coating.

It is worth reporting the guidelines that grade the corrosion resistance that can be offered by any good protective coating. Where the excellent corrosion resistance for a coating can be achieved when a coating has been immersed for several days in a harsh medium like 3.5% NaCl solution, this coating has a value of R_P2_ > 108 Ω cm^2^. In our case, the R_P2_ values are all higher than this recommended value, which indicates that our coating offers excellent corrosion resistance and is sufficiently protective. This is because any decrement in the values of R_P2_ below the mentioned limit provides confirmation of the failure of the coating and consequent formation of a corrosion product below the coating [43,44,45]. Q1 is considered as a double layer capacitance (CPEdl) with some pores because their “*n*” values are very close to unity (1 > *n* > 0.6). Furthermore, Q2, with its “*n*” values around 0.0 for most tested samples (1 > *n* > 0), is probably a coating inductance (CPEI) giving the surface more passivation to the surface.

It can be seen in Figure 9 that the E-1 coating has the highest diameter of the obtained semicircle. On the other hand, the change in hardener balance for E-2 coating decreases the corrosion resistance of the epoxy coating. This indicates that the E-1 coating sample is ideal for increasing the corrosion resistance of the epoxy coating. Prolonging the immersion time to 7 days (Figure 10) shows almost the same behavior for all coatings, but with lower values of both Z′ and −Z″. This effect of decreasing the plotted values of Z′ and −Z″ is increased with the increase of the immersion time (to 14 days, 21 days, and 30 days), which indicates that the increase of immersion time may increase the degradation of the coatings. The Nyquist plots for E-1 and E-2 coating samples also reflected that the corrosion resistance of the coatings is deteriorated by generating failure sites on the coating surface after long-term exposure to the chloride test solution. It has been reported [46] that the failure sites are the small pin holes in the coatings that appear after curing due to solvent evaporation. It has also been reported [47] that these failure sites mostly provide pathways in order for the chloride ions present in the solution to diffuse into the coating, which reduces the resistance for corrosion with extended exposure periods. The Nyquist plots, along with the data of Table 5, revealed that E-1 coating provides the best corrosion resistance in the sodium chloride solution. Also, extending the exposure period decreases the corrosion resistances for all coating formulae due its degradation with time.

Figure 15 shows the Nyquist plots obtained for F-0, F-1, and F-2 after their immersion in 3.5% NaCl solutions for 1 h. The effect of prolonging the exposure periods was also examined as displayed in Figure 16, Figure 17, Figure 18 and Figure 19 for F-0, F-1, and F-2 coatings after 7 days, 14 days, 21 days, and 30 days, respectively. The circuit which was used to fit the obtained Nyquist plots in order to extract the values is represented in Figure 14. The obtained values after fitting the Nyquist plots are listed in Table 6.

It is clear from these figures that F-1 provides the highest corrosion resistance for all lengths of exposure. On the other hand, for the F-0 and F-2 coatings, the corrosion resistance decreases, particularly after long immersion periods. This behavior was also confirmed by the parameters listed in Table 6, where the values of R_S_, R_P1_, and R_P2_ were the highest for the F-1 sample. This effect also lowers the values of Q_1_ and Q_2_; the lowest values of those parameters were recorded also for the F-1 epoxy coating.

According to the listed values for the constant phase elements in Table 6, Q_1_ and Q_2_ can be considered as double layer capacitors. It is well known that the value of the “*n*” component that accompanies the first constant phase elements, Q1, for the current samples varies in the range, 0.99 ≥ *n* ≥ 0.97 (i.e., close to 1), while its value in the case of Q_2_ varies in a wide range of 0.9 > *n* > 0.0. The epoxy coatings are thus very resistive even if they have porosities when their surfaces form a corrosion product layer. In this case, the dissolution of the coating will take place through the diffusion from the porous corrosion product layer. Thus, the degradation of the epoxy formulae, F-0, F-1, and F-2, during their immersion in chloride ions takes place via diffusion. The semicircle of the impedance plot is ideal when the value of “*n*” is very close to one. Moreover, the values of “*n*” that accompany Q_2_ vary with coating and with exposure time to one another, giving a depressed semicircle, and the circuit (Figure 14) gives a real capacitance. In addition to the increase of all solution and polarization resistances, the decrease of the Y_Q1_ and Y_Q2_ values for F-1 coating is due to its excellent corrosion resistance. This confirms that this formula, F-1, has the best corrosion passivation in the chloride solution even after prolonging the exposure time to 30 days. As compared to previous studies [27,28], the current coating formulae provide excellent corrosion protection in the test 3.5% NaCl solutions even after 30 days’ immersion.

## 4. Conclusions

The coating formulations were prepared using two types of hardeners, namely PA-450 and PA-3282. The properties of the coatings were analyzed for both the hardeners under different stoichiometric balances. The coatings were subjected to 7 days of curing, and then, analyzed to obtain mechanical and electrochemical properties. The results revealed that PA-450-based epoxy coating possess slightly better mechanical properties than that of PA-3282-based coating. The final decision of coating performance was made based on electrochemical properties and variation in hardeners led to serious differences in the anticorrosion performance. PA-450 possesses far superior properties compared to PA-3282 even after exposure to 30 days in NaCl solution. With slightly higher mechanical properties and better anticorrosion properties, PA-450 along with filler addition might act as the best candidate in enhancing mechanical and anticorrosion properties.

## Figures and Tables

**Figure 1 materials-15-02562-f001:**
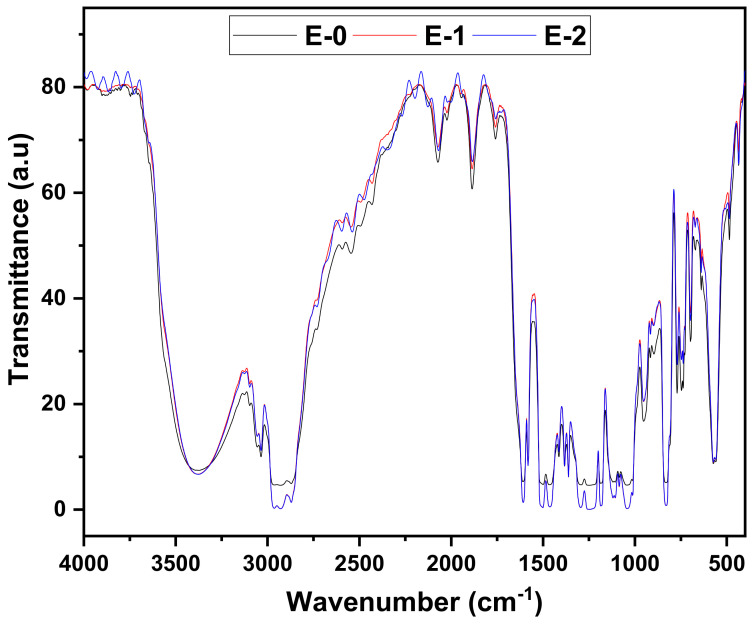
FTIR spectrum of formulation with PA-450 hardener.

**Figure 2 materials-15-02562-f002:**
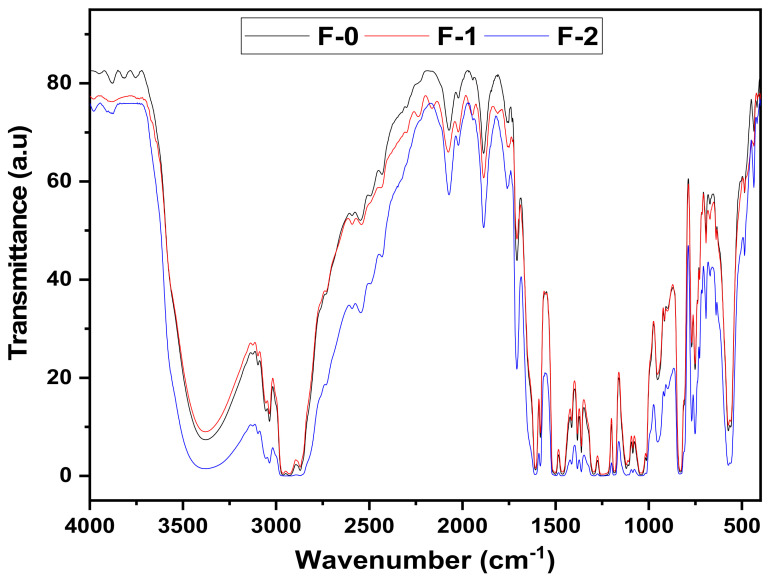
FTIR spectrum of formulation with PA-3282 hardener.

**Figure 3 materials-15-02562-f003:**
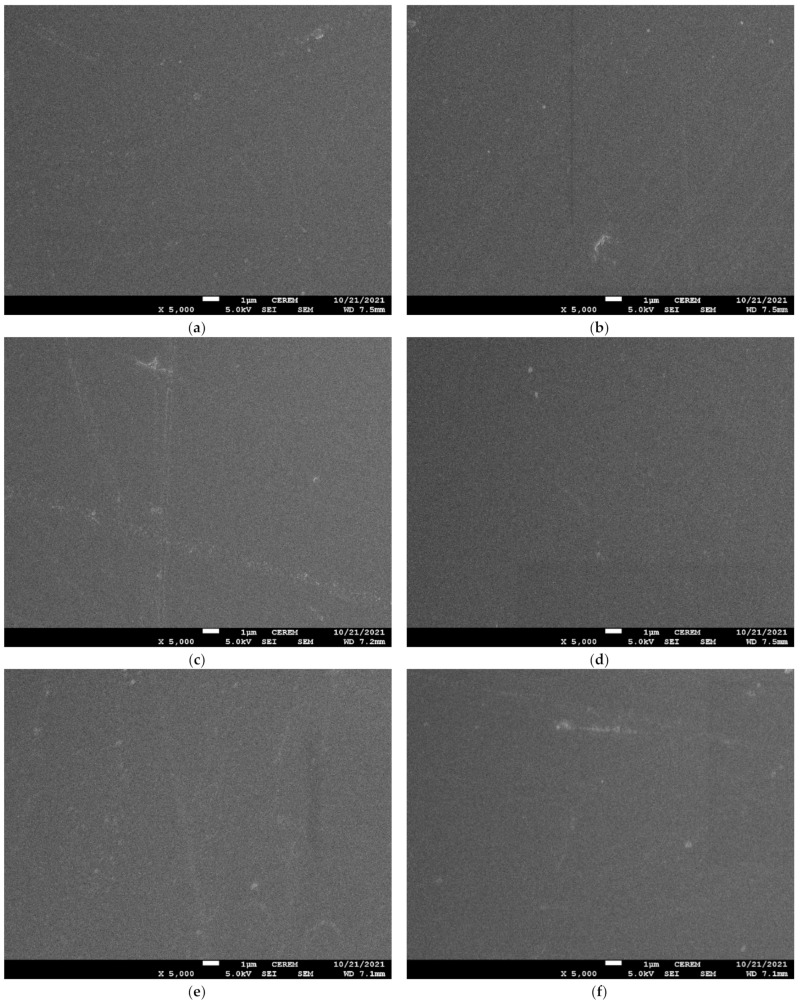
SEM images of prepared formulations with variation in hardener percentages. (**a**) E-0, (**b**) E-1, (**c**) E-2, (**d**) F-0, (**e**) F-1, (**f**) F-2.

**Figure 4 materials-15-02562-f004:**
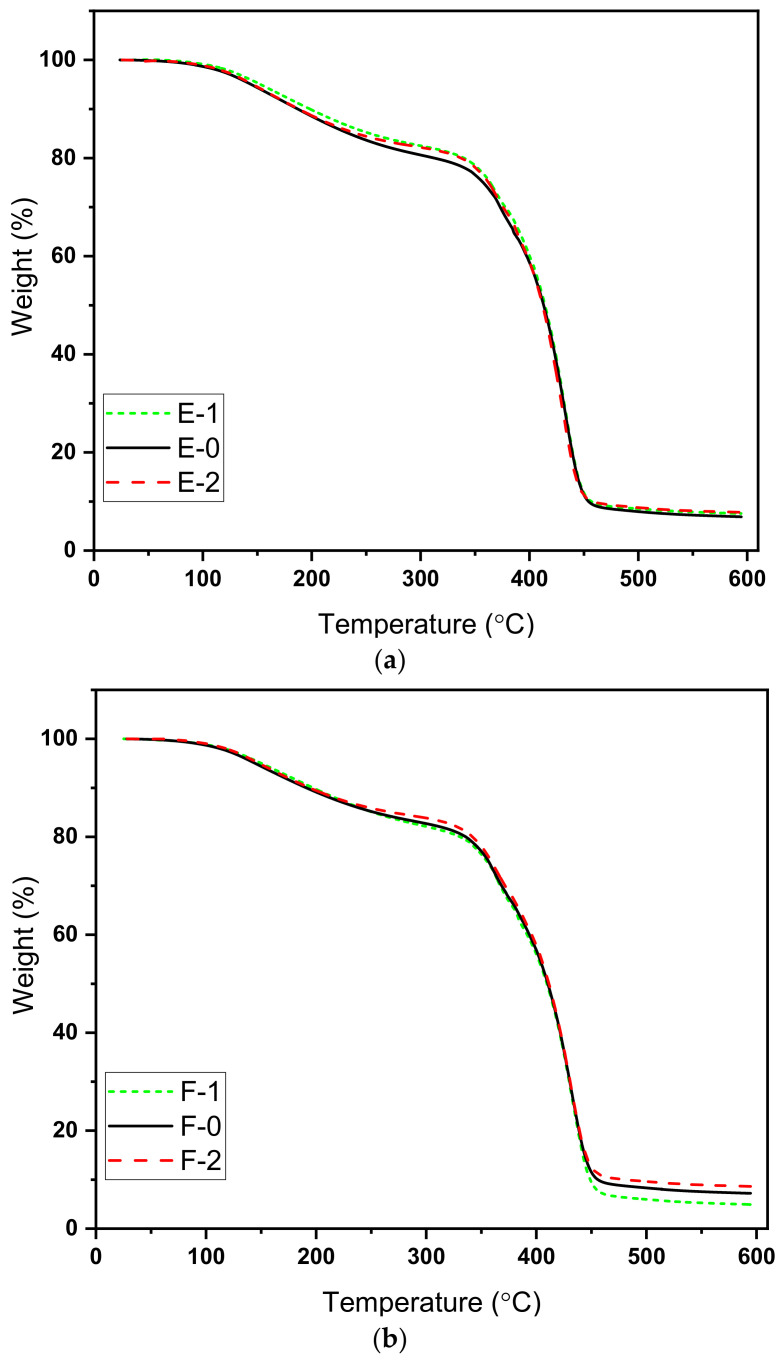
TGA curves for prepared samples (**a**) Hardener 450-BD with variation in hardener percentage (**b**) Hardener 3282-1 BD with variation in hardener percentage.

**Figure 5 materials-15-02562-f005:**
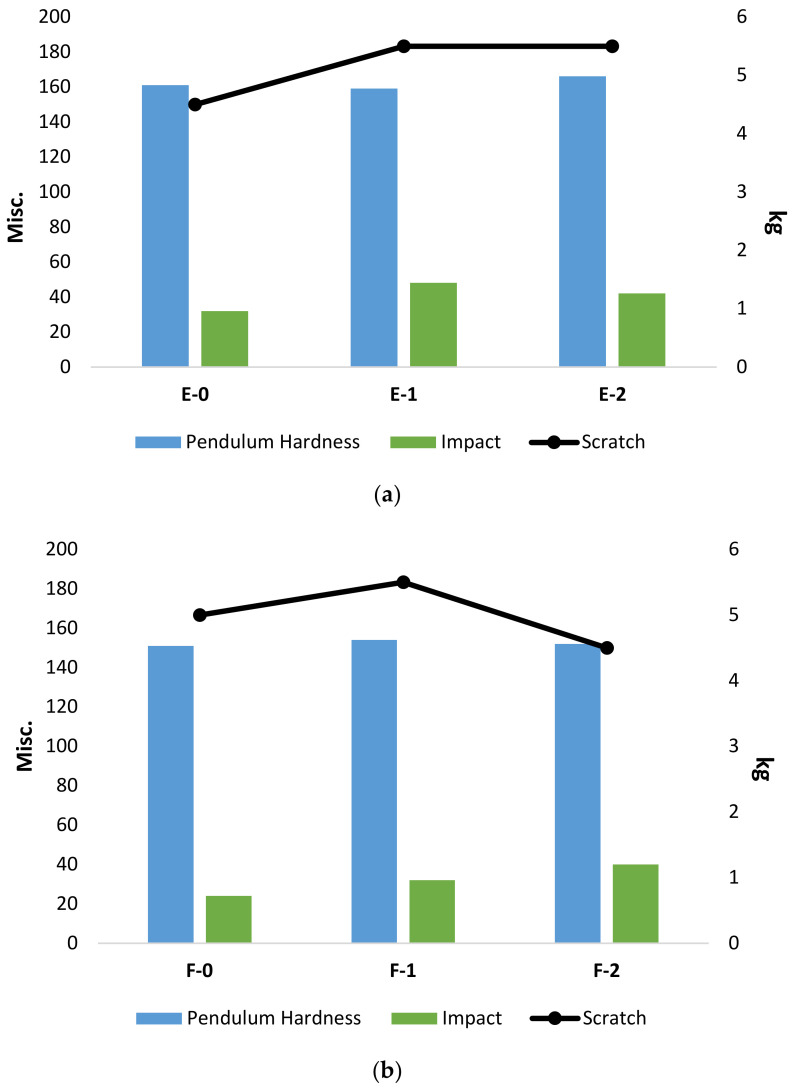
Graphical Representation of obtained mechanical properties (**a**) Coatings prepared with PA-450, (**b**) Coatings prepared with PA-3282.

**Figure 6 materials-15-02562-f006:**
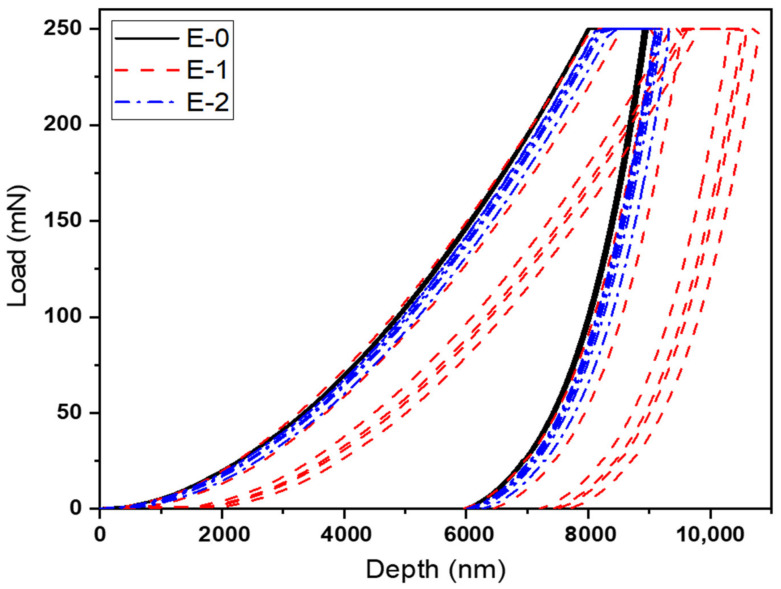
Load vs. depth curve for coatings prepared with PA-450 at different percentages.

**Figure 7 materials-15-02562-f007:**
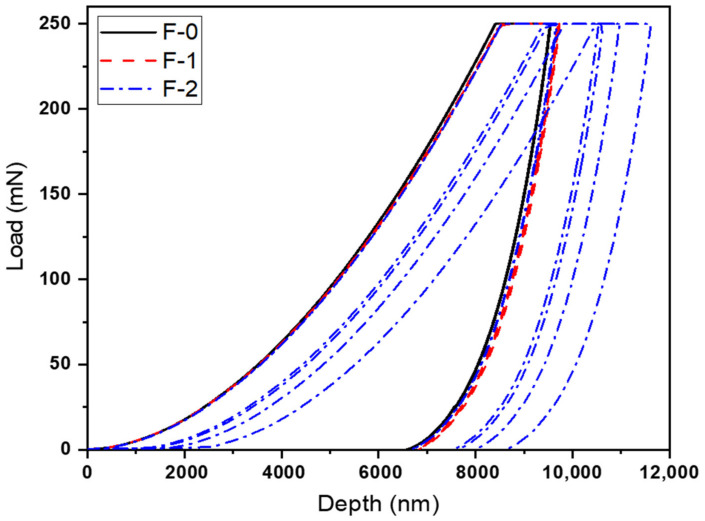
Load vs. depth curve for coatings prepared with PA-3282 at different percentages.

**Figure 8 materials-15-02562-f008:**
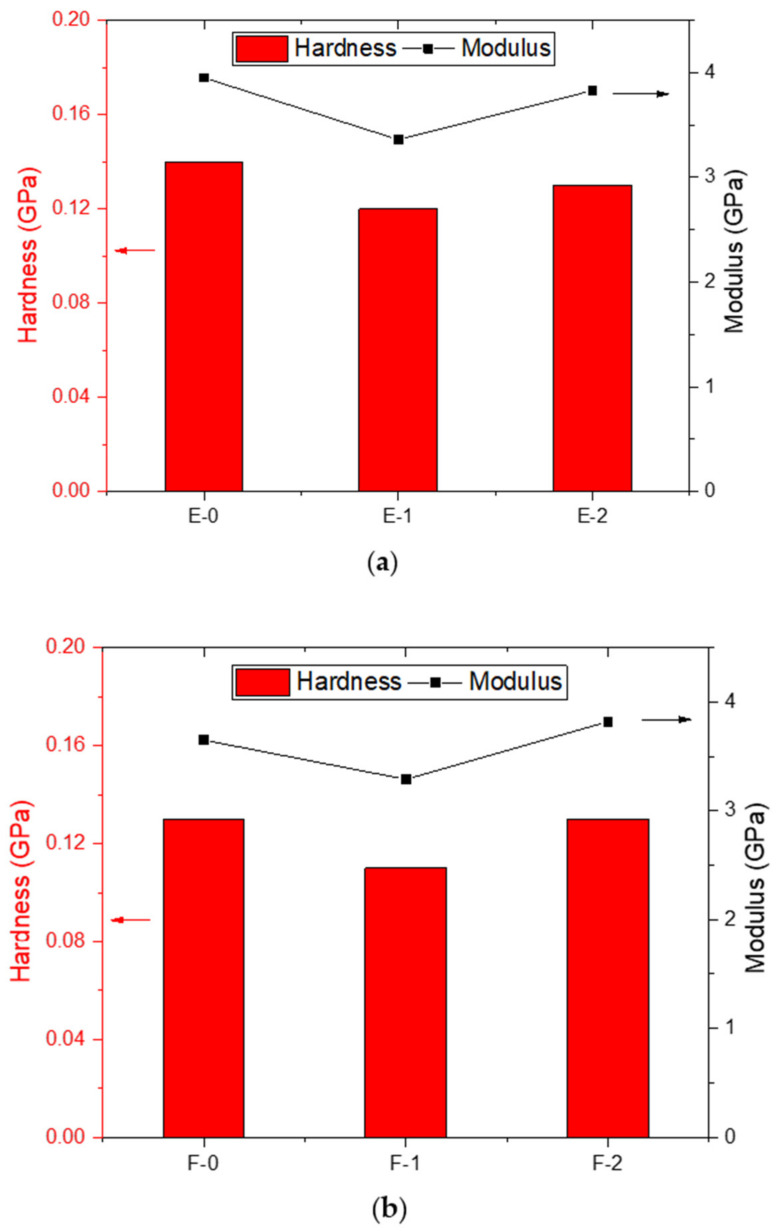
Graphical representation of nanoin2dentation results (**a**) Coatings prepared with PA-450 (**b**) Coatings prepared with PA-3282.

**Figure 9 materials-15-02562-f009:**
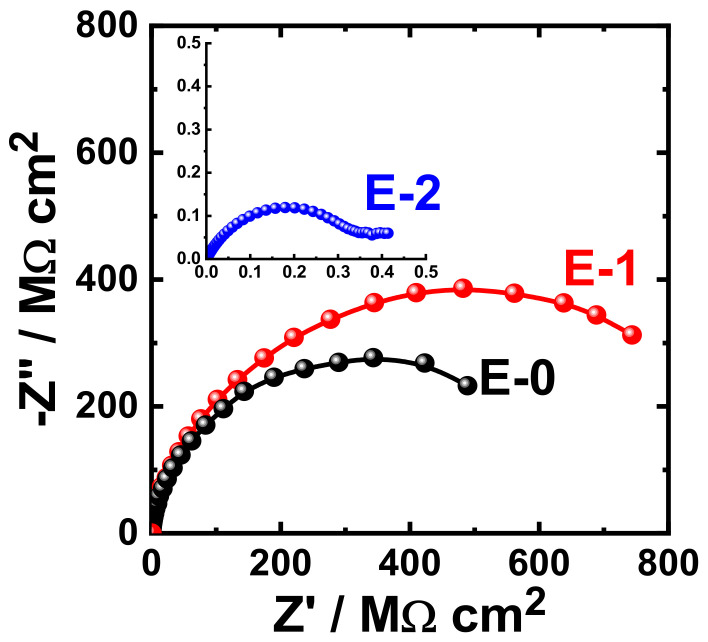
Typical Nyquist plots obtained for E-0, E-1, and E-2 after their immersion in 3.5% NaCl solutions for 1 h.

**Figure 10 materials-15-02562-f010:**
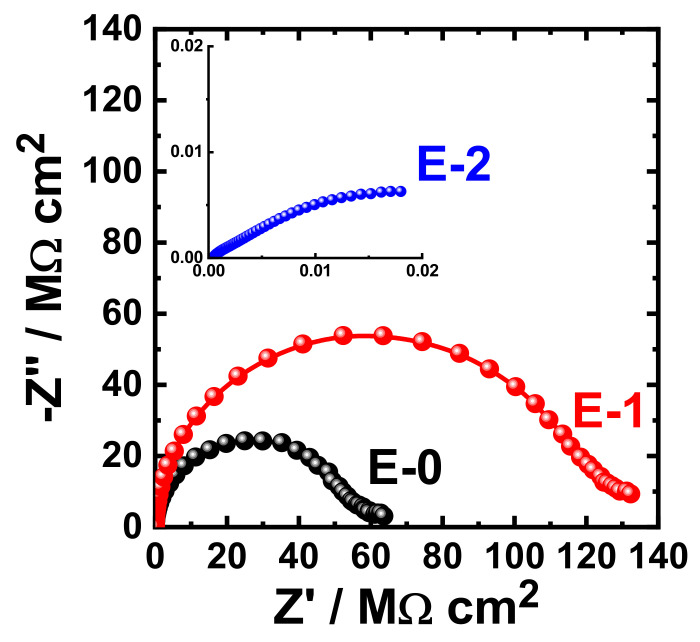
Typical Nyquist plots obtained for E-0, E-1, and E-2 after their immersion in 3.5% NaCl solutions for 7 days.

**Figure 11 materials-15-02562-f011:**
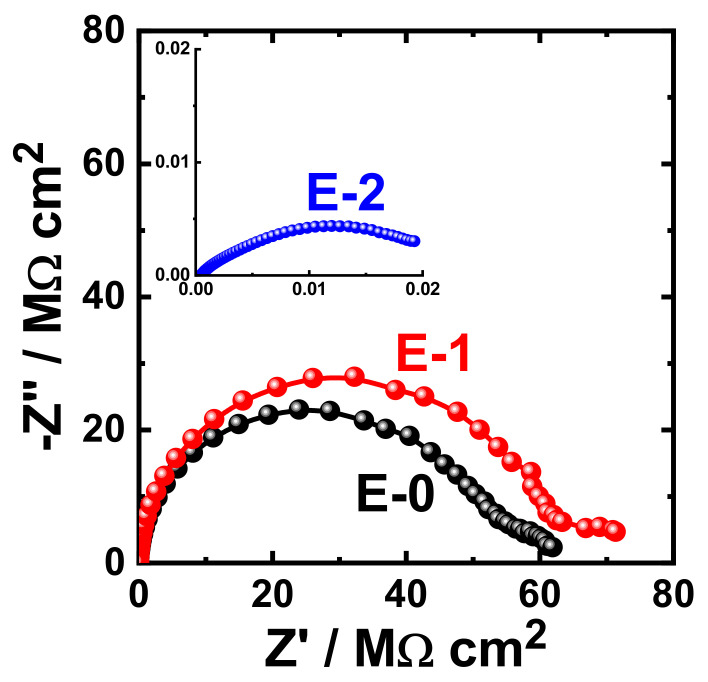
Typical Nyquist plots obtained for E-0, E-1, and E-2 after their immersion in 3.5% NaCl solutions for 14 days.

**Figure 12 materials-15-02562-f012:**
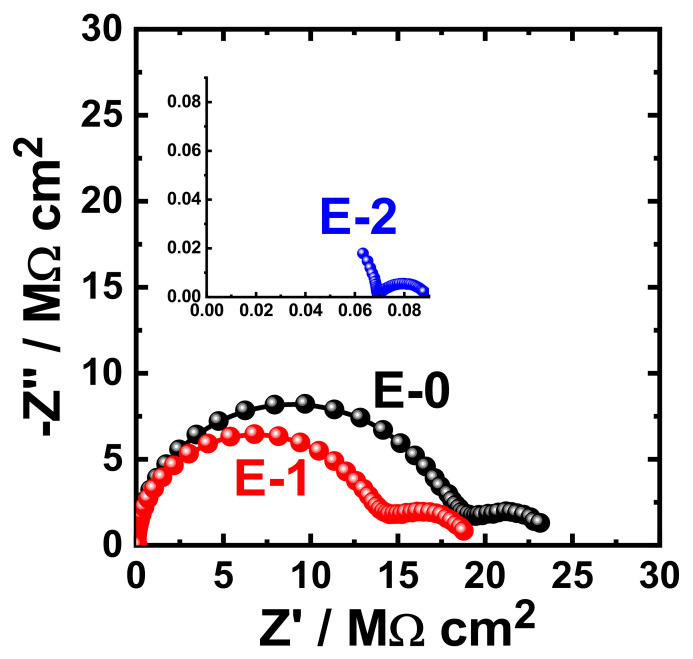
Typical Nyquist plots obtained for E-0, E-1, and E-2 after their immersion in 3.5% NaCl solutions for 21 days.

**Figure 13 materials-15-02562-f013:**
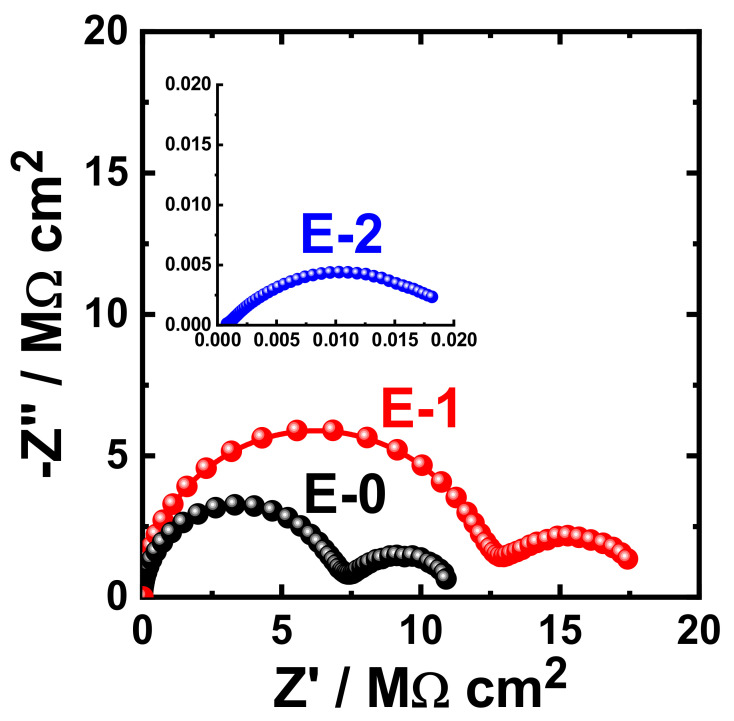
Typical Nyquist plots obtained for E-0, E-1, and E-2 after their immersion in 3.5% NaCl solutions for 30 days.

**Figure 14 materials-15-02562-f014:**
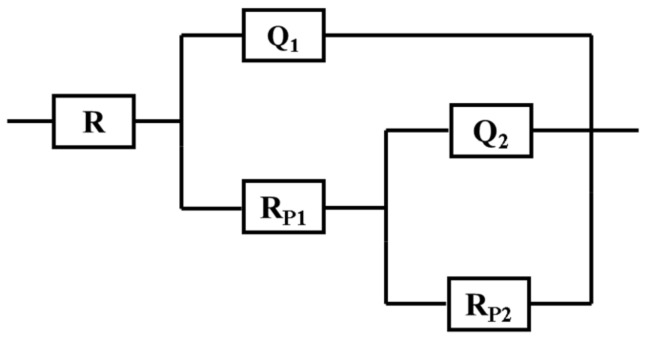
Circuit used to fit the Nyquist plots. (R_S_) is solution resistance, (R_P1_) the polarization resistance, (Q_1_) the constant phase elements, (R_P2_) another polarization resistance, and (Q_2_) another constant phase element.

**Figure 15 materials-15-02562-f015:**
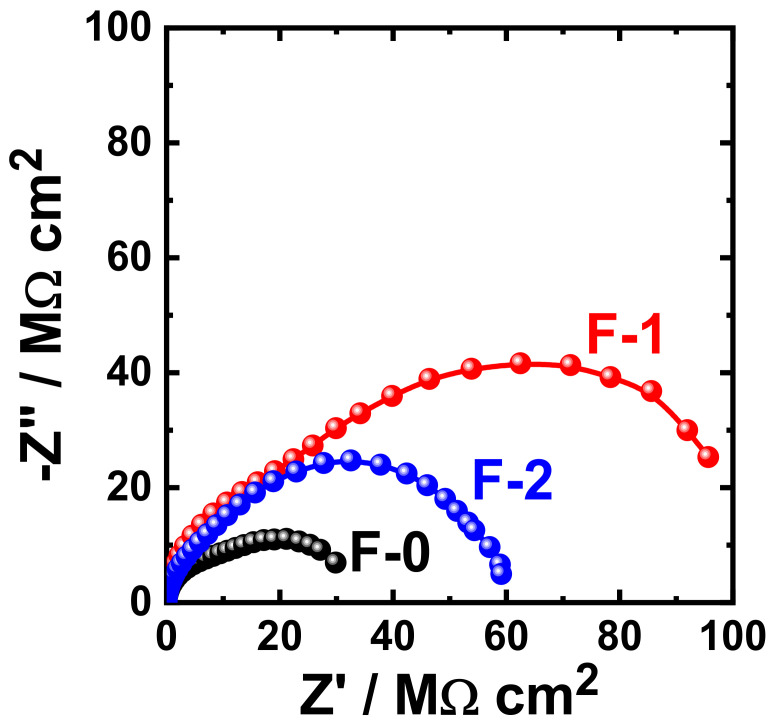
Typical Nyquist plots obtained for F-0, F-1, and F-2 after their immersion in 3.5% NaCl solutions for 1 h.

**Figure 16 materials-15-02562-f016:**
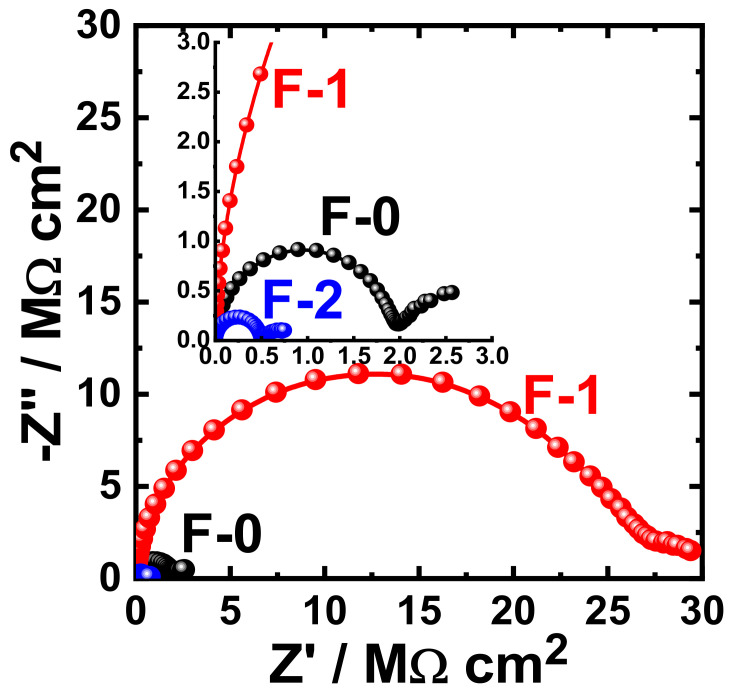
Typical Nyquist plots obtained for F-0, F-1, and F-2 after their immersion in 3.5% NaCl solutions for 7 days.

**Figure 17 materials-15-02562-f017:**
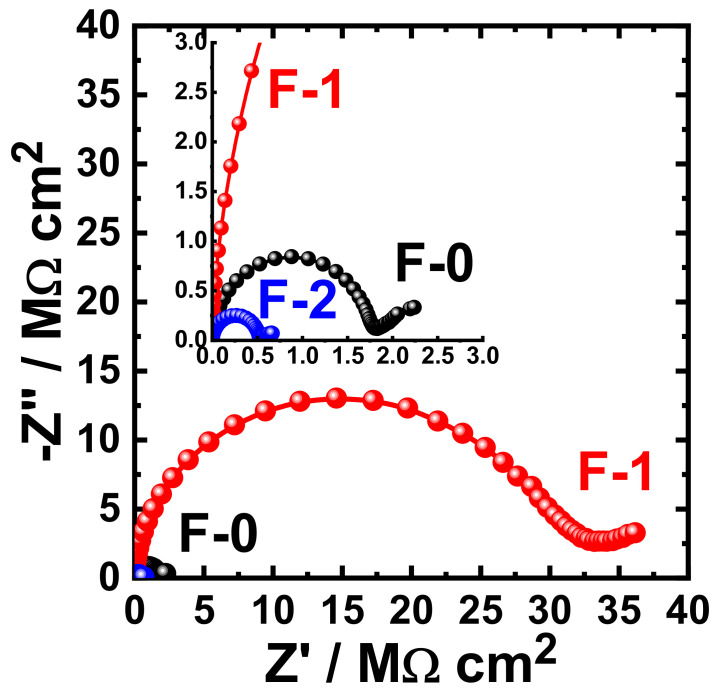
Typical Nyquist plots obtained for F-0, F-1, and F-2 after their immersion in 3.5% NaCl solutions for 14 days.

**Figure 18 materials-15-02562-f018:**
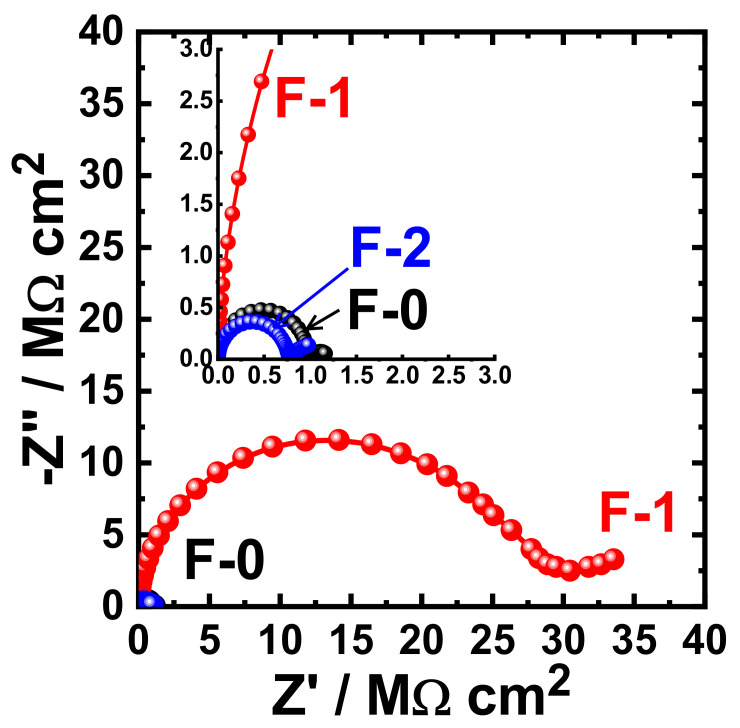
Typical Nyquist plots obtained for F-0, F-1, and F-2 after their immersion in 3.5% NaCl solutions for 21 days.

**Figure 19 materials-15-02562-f019:**
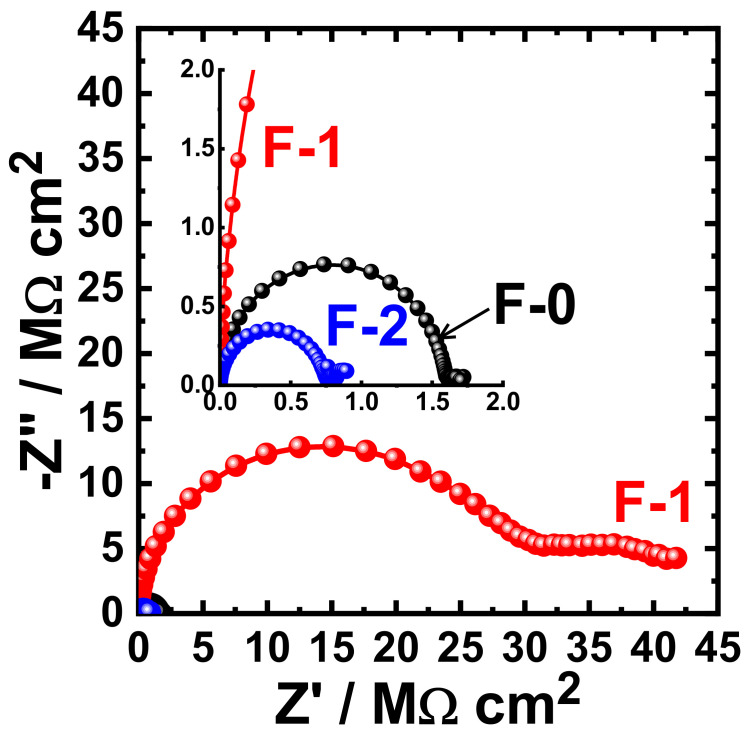
Typical Nyquist plots obtained for F-0, F-1, and F-2 after their immersion in 3.5% NaCl solutions for 30 days.

**Table 1 materials-15-02562-t001:** Formulating ingredients of prepared coatings.

**Formulation Code**	**Hardener PA-450**	**Xylene**	**MIBK**	**Stoichiometry**
E-0	16.66	10	10	Balanced (+10)
E-1	15.90	10	10	+5
E-2	17.50	10	10	+15
**Formulation Code**	**Hardener PA-3282**	**Xylene**	**MIBK**	**Stoichiometry**
F-0	16.34	10	10	Balanced (−10)
F-1	15.52	10	10	−5
F-2	17.15	10	10	−15

**Table 2 materials-15-02562-t002:** Degradation temperatures at different weight loss percentages.

Sample	T15% (°C)	T25% (°C)	T50% (°C)	T75% (°C)	Residue (%)
E-0	253.59	363.98	414.35	435.76	7.55
E-1	240.22	360.54	413.03	435.24	7.88
E-2	241.19	362.48	412.04	434.28	7.78
F-1	251.55	355.04	409.59	433.82	7.92
F-2	252.96	356.59	410.47	434.48	7.20
F-3	268.70	359.06	411.47	434.83	7.61

**Table 3 materials-15-02562-t003:** Pendulum hardness, Scratch and Impact Resistance obtained for PA-450 and PA-3282.

Sample	Pendulum Hardness (Oscillations)	Scratch (kg)	Impact (lb/in^2^)
E-0	161	4.5	32
E-1	159	5.5	48
E-2	166	5.5	42
F-1	151	4	24
F-2	154	5.5	32
F-3	152	4.5	40

**Table 4 materials-15-02562-t004:** Obtained values of Hardness and Modulus for coatings prepared with PA-450 and PA-3282 in different percentages.

Sample	Hardness (GPa)	Modulus (GPa)
E-0	0.14	3.95
E-1	0.12	3.36
E-2	0.13	3.83
F-1	0.13	3.65
F-2	0.11	3.29
F-3	0.13	3.82

**Table 5 materials-15-02562-t005:** Electrochemical impedance spectroscopy (EIS) parameters obtained by fitting the Nyquist plots for coating containing Aradur 450.

Sample	Parameters						
	R_S_/ Ω	Q1		R_P1_/ MΩ	Q2		R_P2_/ MΩ
		Y_Q1_/pMΩ	n		Y_Q2_/nMΩ	n	
E-0 (1 h)	612	0.0316	0.97	27,600	3.34	0.001	296
E-1 (1 h)	440	0.00165	0.97	99,000	3.30	0.001	293
E-2 (1 h)	371	0.767	0.72	0.374	0.00120	0.94	0.0094
E-0 (7 days)	354	961	0.98	248	2.30	0.027	133
E-1 (7 days)	108	0.00141	0.99	432	1.07	0.051	235
E-2 (7 days)	449	0.00473	0.68	0.00160	0.00495	0.09	0.0034
E-0 (14 days)	334	972	0.99	437	2.3	0.073	350
E-1 (14 days)	136	0.00145	0.99	852	1.90	0.002	316
E-2 (14 days)	478	0.00136	0.60	0.0044	0.00436	0.08	0.0018
E-0 (21 days)	846	0.0109	0.97	174	251	0.06	6.72
E-1 (21 days)	262	1540	0.98	13.1	251	0.60	6.82
E-2 (21 days)	196	84,100	0.97	0.0071	0.00121	0.62	0.0021
E-0 (30 days)	620	0.00106	0.98	6.85	396	0.64	5.02
E-1 (30 days)	245	0.00151	0.98	12.2	424	0.69	6.69
E-2 (30 days)	780	0.0044	0.52	0.0066	89.6	0.58	0.0046

**Table 6 materials-15-02562-t006:** EIS parameters obtained by fitting the Nyquist plots for coating containing PA-3282.

Sample	Parameters						
	R_S_/ Ω	Q1		R_P1_/ MΩ	Q2		R_P2_/ MΩ
		Y_Q1_/pMΩ	n		Y_Q2_/nMΩ	n	
F-0 (1 h)	318	943	0.98	12.7	7.78	0.80	21.9
F-1 (1 h)	352	979	0.98	35.3	4.09	0.88	75.7
F-2 (1 h)	285	1.04	0.98	18.9	2.91	0.85	41.6
F-0 (7 days)	424	1.02	0.98	1.91	0.00119	0.63	2.02
F-1 (7 days)	116	954	0.99	38.4	1.17	0.21	71.0
F-2 (7 days)	273	1.12	0.98	0.00494	0.00908	0.57	0.0043
F-0 (14 days)	315	0.00103	0.98	1.73	0.00235	0.48	2.69
F-1 (14 days)	292	991	0.98	193	387	0.08	247
F-2 (14 days)	246	1.13	0.97	0.00523	0.00168	0.48	0.0045
F-0 (21 days)	259	963	0.98	0.00982	0.00110	0.60	0.0027
F-1 (21 days)	346	991	0.98	67.5	5.62	0.03	21.5
F-2 (21 days)	210	1.11	0.98	0.00758	0.00117	0.49	0.0095
F-0 (30 days)	205	944	0.99	1.57	0.00709	0.41	0.0024
F-1 (30 days)	312	981	0.98	2.16	52.7	0.33	3.44
F-2 (30 days)	446	1.15	0.97	0.00752	0.00148	0.55	0.0029

## Data Availability

All the data is available within the manuscript.

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
