# Peer review of "Development and Characterization of PA 450 and PA 3282 Epoxy Coatings as Anti-Corrosion Materials for Offshore Applications"

_materials, 2022, doi:10.3390/ma15072562_

Round 1

Reviewer 1 Report

The manuscript (materials-1621971) presents the use of two types of hardeners a polyaminoamine (Aradur 450 BD) and a polyamidoamine (Aradur 3282 BD), at three different stoichiometries, for the fabrication of epoxy coatings. The obtained films were characterized by SEM, FTIR, TGA, and mechanical characterization (nanoindentation techniques). Also, the electrochemical corrosion behavior was investigated using electrochemical impedance spectroscopy. The manuscript is logically arranged and well structured, however, the novelty of the study is rather low. Even the optimization process cannot be properly addressed with only three experiments being performed – an experiment design should be developed and interpreted with respect to the final coating characteristics (this can be achieved using specialized software such as Statistica or Design-Expert).

Author Response

file attached

Reviewer 2 Report

The paper entitled "Developing and characterization of PA 450 and PA 3282 epoxy coatings as anticorrosion materials for offshore applications" presents the interesting study focused on the anticorrosion coating optimization. Taking into account the novelty aspect, the research is not new, however, the presented technique has a great implementation potential. Before publication few additional aspects should be taken into account.

-the structure analysis (SEM) is not showing any visible difference between the prepared coatings, I suggest to prepare the images with higher focus, or perform AFM measurements

-the results of the mechanical tests should be presented in the graphical way, samples, especially for scrach resistance tests, the appearance of the residual coating should be examinated by SEM microscopy

Reviewer 3 Report

In its current form, the manuscript requires major corrections. Some of the suggestions are as follows:

  1. I don't think it's the best solution to use abbreviations in keywords.
  2. The Introduction section needs to be significantly updated and expanded. for each previous study individually, it is necessary to state what the authors did and what is the most important result.
  3. Then it is necessary to point out the common characteristics of previous research.
  4. It is then necessary to highlight the shortcomings of previous research.
  5. Then you need to define the goal of your research and scientific hypothesis.
  6. Finally, it is necessary to emphasize the scientific contribution of your research and its innovation.
  7. Why is it important to point out in scientific work, since when are hardeners purchased?
  8. Why 7 days? Why 1 hour? Why 30 days? Why is this the optimal number of hours/days? Further elaborate in the manuscript.
  9. The discussion of the obtained results must be much, more extensive, comprehensive and deeper. Process physics must be scientifically discussed. The results obtained should also be compared with similar previously published research.
  10. Analyse and discuss potential errors. Perform sensitivity analysis and / or uncertainty analysis of results.
  11. What is the accuracy of the obtained results, or even better, estimate the measurement uncertainty of your results.
  12. Analyse and discuss the possibilities of applying your methodology in practice.
  13. In the conclusions, state the limitations of your methodology and future research.

Round 2

Reviewer 3 Report

Great job. Excellent research. The article has been updated and corrected. I am glad that the authors understood that all the suggestions were well-intentioned. He suggests accepting the article in its current form.